# Repeated Transcranial Photobiomodulation with Light-Emitting Diodes Improves Psychomotor Vigilance and EEG Networks of the Human Brain

**DOI:** 10.3390/bioengineering10091043

**Published:** 2023-09-05

**Authors:** Akhil Chaudhari, Xinlong Wang, Anqi Wu, Hanli Liu

**Affiliations:** Department of Bioengineering, University of Texas at Arlington, 500 UTA Blvd, Arlington, TX 76019, USA; akhilmchaudhari@gmail.com (A.C.); xl.victor.wang@gmail.com (X.W.); anqi.wu@uta.edu (A.W.)

**Keywords:** transcranial photobiomodulation, tPBM, repeated tPBM, light-emitting diodes, psychomotor vigilance task, PVT, electroencephalography, EEG

## Abstract

Transcranial photobiomodulation (tPBM) has been suggested as a non-invasive neuromodulation tool. The repetitive administration of light-emitting diode (LED)-based tPBM for several weeks significantly improves human cognition. To understand the electrophysiological effects of LED-tPBM on the human brain, we investigated alterations by repeated tPBM in vigilance performance and brain networks using electroencephalography (EEG) in healthy participants. Active and sham LED-based tPBM were administered to the right forehead of young participants twice a week for four weeks. The participants performed a psychomotor vigilance task (PVT) during each tPBM/sham experiment. A 64-electrode EEG system recorded electrophysiological signals from each participant during the first and last visits in a 4-week study. Topographical maps of the EEG power enhanced by tPBM were statistically compared for the repeated tPBM effect. A new data processing framework combining the group’s singular value decomposition (gSVD) with eLORETA was implemented to identify EEG brain networks. The reaction time of the PVT in the tPBM-treated group was significantly improved over four weeks compared to that in the sham group. We observed acute increases in EEG delta and alpha powers during a 10 min LED-tPBM while the participants performed the PVT task. We also found that the theta, beta, and gamma EEG powers significantly increased overall after four weeks of LED-tPBM. Combining gSVD with eLORETA enabled us to identify EEG brain networks and the corresponding network power changes by repeated 4-week tPBM. This study clearly demonstrated that a 4-week prefrontal LED-tPBM can neuromodulate several key EEG networks, implying a possible causal effect between modulated brain networks and improved psychomotor vigilance outcomes.

## 1. Introduction

Transcranial photobiomodulation (tPBM) is a new, noninvasive intervention tool that involves delivering near-infrared (NIR) light in the range of 800–1070 nm with laser or light-emitting diodes (LEDs) to the human head [1,2,3]. This type of light can safely reach the cortical and subcortical regions, as well as stimulate neuronal function as a reparative and protective tool in a noninvasive manner [4,5,6]. The underlying principle of tPBM is based on mitochondrial light absorption by cytochrome c oxidase (CCO), which plays a key role in neuronal oxygen utilization for energy metabolism [3,7,8]. This photonics–bioenergetics mechanism results in unique metabolic effects on the living brain, with benefits for neural enhancement and neuroprotection [9,10]. Numerous recent studies have reported that tPBM has facilitated the effective treatment of neurological disorders [11,12,13,14,15,16], including Alzheimer’s disease (AD) and dementia [17,18,19,20,21,22], as reviewed in the literature [23,24,25]. The benefits of tPBM as an intervention strategy include its low cost, compactness, self-administration, and routine use in repeated care.

For tPBM to become an effective therapeutic tool, it is necessary to understand its underlying neurophysiological mechanisms. It is accepted that the oxidized form of CCO (oxi-CCO) plays a key role in the utilization of neuronal oxygen for energy metabolism [11]. This CCO-driven mechanism of tPBM was experimentally demonstrated by Wang et al. [26,27,28], who showed that tPBM at 1064 nm can noninvasively stimulate mitochondrial metabolism in living human tissues. This set of experimental observations is supported by reproducible results [29] and another independent human study [30].

However, studies on tPBM-induced electrophysiological effects in the human brain are limited, with only a few publications [31,32,33,34] besides ours [26,35,36] in the last 4–5 years. All these studies reported acute alterations in electroencephalogram (EEG) power by tPBM, without any repeated tPBM treatment, compared to sham stimulation. Furthermore, the protocol used to achieve significant improvement in cognition in older adults with or without AD required daily tPBM administration over one or two months, respectively [18,19]. Such critical daily administration of tPBM appears to be the key to success and thus necessitates an understanding of the repeated effects of tPBM on the human brain. In addition, while a few articles have reported cognitive improvements by acute tPBM in healthy humans [37,38], no report has been found in the literature on repeated vigilance improvement along with their electrophysiological effects after repeated tPBM administration for weeks in healthy participants.

The psychomotor vigilance task (PVT) is a sustained attention game that has been used to measure the speed of response to a visual stimulus, and it has been recognized as an effective tool for evaluating sustained attentional performance [39]. Thus, PVT has been routinely used to examine and demonstrate significant acute improvements in attention after laser stimulation [37,38,40]. Vargas et al. also showed an improvement in the reaction time over five weeks under laser stimulation in older adults [32].

As a proof-of-principle methodology development study, we designed and conducted the first sham-controlled human study to demonstrate that 4-week repeated tPBM via a cluster of LEDs (LED-tPBM) enabled a significant improvement in human vigilance performance in healthy young adults. Moreover, we recorded acute and repeated electrophysiological effects using a 64-channel EEG during the first and last sessions of the 4-week tPBM or sham experiments. Consequently, we proposed three hypotheses that would be be proven at the end of this study. First, repeated LED-tPBM of the right forehead of healthy subjects significantly improved vigilance. Second, acute LED-tPBM significantly increased topographical EEG delta and alpha powers. Third, the four-week repeated LED-tPBM enhanced several brain network strengths (such as default mode, executive control, and frontal parietal networks) in all five EEG frequency bands. Specifically, we utilized frequency-domain analysis to quantify channel-wise EEG power alterations by acute LED-tPBM to support Hypothesis 2 [41], and a newly developed inverse algorithm [36] to demonstrate that repeated tPBM significantly enhanced key EEG brain networks to support Hypothesis 3. In conclusion, this study revealed an objective association between improved vigilance and tPBM-modulated brain networks in healthy humans, providing evidence that tPBM is a potential intervention tool for the enhancement of human cognition in healthy humans, and thus has great potential for the effective treatment of neurological diseases, including AD.

## 2. Materials and Methods

### 2.1. Study Participants

This study employed a between-group experimental design. Twenty-two healthy volunteers (7 males and 15 females; mean age = 26 ± 5 years) were recruited from the university community. Each subject was randomly assigned to either the active or sham group; therefore, there were 11 participants in each of the tPBM and sham groups. All participants had normal or corrected-to-normal vision, and were neurologically and psychologically healthy. Subjects were excluded if they were younger than 18 years of age, currently pregnant (during the recruitment), had a neurological disease, diabetes, psychiatric illness, had any tattoos on the head and face, or had consumed caffeine within 3 h before the experiment (PVT is believed to be sensitive to caffeine [42]). The study protocol complied with all applicable federal guidelines and was approved by the Institutional Review Board (IRB) of the University of Texas at Arlington. Written informed consent for the sham-controlled LED-tPBM experiment was obtained from each participant before each set of experiments. All participants were compensated for their participation after their last visit.

### 2.2. Experimental Setup and Protocol for LED-tPBM

LED-tPBM stimulation was delivered to the right forehead using a continuous-wave (CW) LED handheld probe (Thor-LX2, THOR Photomedicine Inc., Amersham, UK; Figure 1a). Table 1 lists several key parameters of the LED-tPBM setup. Accordingly, the average irradiance per wavelength was 30 mW/cm^2^; thus, the maximal optical dose or fluence delivered to the human forehead by tPBM over 10 min was approximately equal to =30 mJ/s/cm^2^ × 32 cm^2^ × 10 min × 60 s/min = 576 J. On the other hand, the power used for the sham group was set to 0.0 W.

In both the active and sham groups, the participants were seated comfortably while keeping their heads upright and wearing a pair of safety goggles (Figure 1b,c). Since the LEDs had low irradiance, the participants wearing goggles were unable to “see” or “feel” the light, and were thus blind to the active or sham mode while they were told to receive active tPBM during the four-week study.

Figure 1d summarizes the 4-week experimental protocol. It consisted of two sessions separated by 2–3 days per week over four weeks, giving a total of eight sessions—S1 to S8. Each experimental session lasted 20 min, as shown in Figure 1e, consisting of a 2 min baseline and an 8 min trial practice mixed with rest without any LED light exposure. The LED-tPBM was then delivered continuously for 10 min to each participant’s right forehead while they performed two 5 min cycles (i.e., time period 1, TP1, and time period 2, TP2, as marked in Figure 1e) of a 2 min rest epoch and 3 min sustained attention game (PVT). During TP1 and TP2, the reaction times in response to PVT were recorded and used to assess the tPBM-induced improvement in the participants’ vigilance. Note that our data analysis focused only on the 10 min LED-tPBM period for both behavioral and electrophysiological measures to examine the acute effect during S1, as well as the repeated effect between S1 and S8.

### 2.3. Design of PVT and Data Recording of EEG

A PVT designed by Pulsar Informatics was employed [43]. In the PVT, participants were asked to stare at a blank box at the center of a computer monitor and press the spacebar on the computer keyboard as soon as they saw a timer shown on the screen. After pressing, the timer disappeared from the monitor, followed by a returned blank box. The inter-stimulus interval time was approximately one second; each 3 min session of the PVT included 25–30 stimuli per TP. The reaction time for each stimulus was recorded for each participant. The washout period between the two visits within a week was 48 h or longer. Furthermore, for both the sham and active tPBM groups, EEG recordings were taken continuously during the entire 20 min experiment during the first and last sessions (i.e., S1 and S8) using a 64-channel EEG system (Biosemi Instrumentation, Amsterdam, Netherlands) at a data acquisition rate of 512 Hz. Before the experiment started, the participants were asked to calm themselves and pay attention to the game presented before them.

### 2.4. Data Analysis

#### 2.4.1. Behavioral Data Analysis

As shown in Figure 1d, each session included two 3 min epochs of PVT with 25–30 stimuli per TP. The mean performance score for each TP was obtained by averaging the reaction times for all the stimuli per TP. This was repeated for all subjects in both the tPBM and sham groups during each visit, followed by a group-level averaging of the reaction times for both groups. The time-dependent improvement in reaction time over four weeks of sham or active tPBM was evaluated for each group based on Pearson’s correlation coefficient and its statistical significance. In this way, we quantified the repeated effects of LED-tPBM/sham on vigilance and proved our first hypothesis.

#### 2.4.2. EEG Data Preprocessing

The EEG time series were preprocessed using the EEGLAB toolbox [44,45] on the MATLAB platform (R2021b). Raw EEG data taken during the 10 min LED-tPBM or sham period (TP1 and TP2) were initially bandpass-filtered from 0.5 Hz to 70 Hz using a zero-phase Butterworth filter in MATLAB. The power line noise of 60 Hz was removed using a notch filter. The filtered data were then re-referenced to the common means across the 64 electrodes before performing Independent Component Analysis (ICA) to remove the artefacts of saccades, jaw clenches, heartbeats, and eye blinks. ICA was performed using the EEGLAB function “runica”. Components containing artifacts were visually inspected and removed from the data [35,46]. Next, the EEG signals were further segmented into 1 s epochs. An epoch was considered acceptable if the standard deviation of the epoch was in the range of 0.1 μV, i.e., up to 2 times the standard deviation of the channel. Moreover, a threshold of mean ± 4 times the standard deviation of the respective channel was applied to reject the outlier data points within each channel. Six of the participants (four from the tPBM group and two from the sham group) during the first visit, and six subjects during the last visit (three from the tPBM group and three from the sham group) were removed from the study based on the previously mentioned criteria. Accordingly, the EEG data resulting in the observations of this study were based on the measurements from a total of 16 participants.

#### 2.4.3. Global Normalized Power Spectral Density during 10 min tPBM/Sham Epochs

We temporally segmented the preprocessed data into resting and PVT phases as 2 min and 3 min windows, respectively. The power spectrum density (PSD) of each temporal segment was calculated using the native MATLAB function “pwelch” (sampling frequency of 512 Hz, and discrete Fourier transform points of 4 s with 50% overlap) for each electrode and subject, as shown in Step 1 in Figure 2. Pwelch resulted in a 0.25 Hz spectral resolution, ranging from 0 Hz to 128 Hz.

For both rest and PVT epochs in each TP, the PSD values were spectrally averaged in five frequency (5-*f*) bands of interest; namely, delta (0.5–4 Hz), theta (4–7 Hz), alpha (7–13 Hz), beta (13–30 Hz), and gamma (30–70 Hz) bands. This step was followed by normalization with respect to those in the baseline taken at the beginning of the experiment (see Figure 1e). This process produced a group-level normalized PSD (nPSD) at each EEG electrode in each of the five frequency bands. All nPSD values were also averaged across 64 EEG electrodes to obtain a global (or spatial) mean normalized PSD, nPSD_64_, during the rest and PVT in TP1 and TP2 for each participant in both the active and sham groups (Steps 2 and 3 in Figure 2). Furthermore, in TP1 and TP2, all the subjects underwent two resting and two PVT epochs. Paired t-tests were performed on paired nPSD_64_ values between the 2 min rest and the 3 min PVT epochs for each TP1 and TP2 using R studio [47] for both the EEG data taken in S1 and S8 separately, and for each (sham and tPBM) group, in each of 5-*f* bands. This step was to examine whether we could pool the nPSD_64_ values in each frequency band during the rest and PVT epochs together in each of TP1s and TP2s conducted (Step 4 in Figure 2). Significant differences in the nPSD_64_ at each frequency band were set at a significance level of *p* < 0.05.

#### 2.4.4. Topography of the nPSD Alterations at Five Frequency Bands during 10 min LED-tPBMs

To examine the acute effect of a 10 min LED-tPBM, we generated topographic maps of the nPSD at the 5-*f* bands during TP1 and TP2 in S1 of Week 1. Specifically, channel-wise nPSD values at the 5-*f* bands during Session S1 were extracted to obtain group-level averages for both the sham and tPBM groups. These averaged nPSD values were used to generate activation topographical maps for each TP1 and TP2 in Session S1 of Week 1, and this operation was achieved using the EEGLAB toolbox from the MATLAB platform. To statistically compare the differences in the nPSD values topographically between the tPBM and sham groups in each of the 5-*f* bands, we performed cluster-based permutation testing [48] to solve the multi-channel comparisons with the “ft_freqstatistics” function that is available in the Fieldtrip toolbox in MATLAB [49].

The above analysis strategy was repeated to examine the 4-week repeated effects of LED-tPBM on the topographic changes in nPSD between S1 in Week 1 and S8 in Week 4 at all 5-*f* bands, as noted in Step 5 of Figure 2. In this case, cluster-based permutation testing examined whether the 4-week LED-tPBM could create significant alterations in EEG powers at the 5-*f* bands in certain brain regions when compared with those induced by the first 10 min tPBM in Week 1.

#### 2.4.5. The Algorithm to Identify EEG Networks and Their Alterations by Repeated tPBM

To further understand whether repeated LED-tPBM would have any significant impact on brain EEG networks, we applied a newly developed algorithm to identify key networks and their respective alterations induced by a 4-week light intervention. The new algorithm was formulated by combining group singular value decomposition (gSVD) with exact low-resolution brain electromagnetic tomography (eLORETA). The overall process is illustrated in Figure 3.

The gSVD + eLORETA algorithm was recently introduced and published in [36], where a detailed description can be found. For the convenience of readers, a brief summary of the eight steps (as listed in Figure 3) is given below, and more information on each step is provided in Section SA of the Appendix A.

Step 1 involved preprocessing the raw EEG data using the same procedure as described in Section 2.4.2.

Step 2 involved performing a z-score transformation on each preprocessed EEG time series in a selected period to minimize inter-subject variation. This standardization step is necessary for an unbiased operation in gSVD. Group SVD can be considered similar to spatial group independent component analysis (ICA) which has been widely used in the field of fMRI [50,51,52].

Step 3 involved forming the group for computing gSVD based on the standardized EEG time series from all 32 samples of the EEG measurements (n = 7 and 9 participants for the tPBM and sham groups in S1 of Week 1, respectively; n = 8 and 8 for both groups in S8 of Week 4, respectively). All these time series were concatenated into a single 2D matrix, M_gSVD_. One dimension of this matrix was the concatenated time covering TP1 and TP2 in both S1 and S8 for the two groups. The other dimension was 64 channels with standardized EEG readings. There are interconnections between slow and fast EEG rhythms in mediating the brain networks [53]. Hence, the five frequency bands of EEG were considered together, rather than separating them into individual frequency bands while performing gSVD.

Steps 4 and 5 were performed to compute the gSVD across the EEG measures in S1 and S8 for both groups. The concatenated matrix M_gSVD_ was used to identify the common Principal Components (PCs) across the four weeks and in the two groups using the native MATLAB function ‘svd’. At the end of the analysis, we selected 12 gSVD-derived PCs with their respective weights, 1D time series, and corresponding topographies. Consequently, these led to the extraction of 12 gSVD components, which facilitated 2D topographies and 3D source localizations for the 12 EEG brain networks.

Step 6 was performed to quantify the source localization using eLORETA [36], which uses a total of 6239 voxels at a 5-mm spatial resolution. eLORETA offers a weighted least-squares based solution with a localization error [54]. This enabled us to localize the 3D cortical sources of the 2D electric potential distribution of the 12 SVD components. This procedure produced 2D (sagittal, coronal, and axial) views for each brain EEG network. See Section SA of the Appendix A for all the details on the images used for eLORETA.

Step 7 enabled the calculation, achieved using the MATLAB function “pwelch” (with a 20 s window and 50% overlap), of the normalized PSD of the 12 SVD components for the respective time periods, as well as in S1 of Week 1 and S2 of Week 4 for both groups. We used a 4 s window length with a 50% overlap in our PSD calculations for regular EEG signals (see Section 2.4.3). However, we chose a 20 s window length to quantify the PSD for each SVD component because we wished to have a balance between a smooth PSD and adequate frequency details.

Step 8 was required to compute the power changes in the EEG networks induced by the 4-week LED-tPBM. First, we obtained the spectrally averaged PSD power for each subject for each of the 5-f bands during TP1 and TP2 in Sessions S1 and S8, followed by baseline normalization (nPSD) for all 12 SVD components in both the tPBM and sham groups, as well as for TP1 and TP2. Furthermore, we computed the differences in the normalized network powers (nnp) between S8 in Week 4 and S1 in Week 1 (i.e., ΔnP = nnp_S8 − nnp_S1) for each of the sham and tPBM groups in all 5-f bands during TP1 and TP2, respectively. Finally, the significant differences of ΔnP between the tPBM versus sham groups were determined by performing two-sample, non-parametric tests [55,56] during TP1 and TP2 for each component in each frequency band at the significance level of *p* < 0.05 (marked by “*”) and *p* < 0.01 (marked by “&”). A MATLAB function of “ranksum” was used for the non-parametric permutation comparisons between the LED and sham groups in the 5 frequency bands and the 12 brain EEG networks.

## 3. Results

### 3.1. Repeated 4-Week LED-tPBM Significantly Improves Psychomotor Vigilance

As the acute effect of tPBM on psychomotor vigilance performance may not be significant, repeated or longitudinal interventions over a period of weeks were needed to achieve a significant effect of tPBM, as reported in several studies [18,19,57,58]. In our study, the group-averaged reaction times to PVT in both the sham (n = 11) and tPBM (n = 11) groups during a 10 min stimulation over 4 weeks are plotted in Figure 4. Data were fitted using linear regression to observe the trend over time for both groups. Linear fitting showed a significant decrease in the reaction time for the tPBM group (*p* = 0.002; red line), as shown in Figure 4. In contrast, the sham group did not achieve any significant improvement in reaction time over four weeks (*p* =0.07; blue line). This result demonstrates that repeated 10 min LED stimulation twice a week for four weeks improves psychomotor vigilance in healthy, young participants. These results confirmed our first hypothesis. Note that the statistically significant effect of tPBM on the mean reaction time was not affected by an initial bias or offset between the two groups.

As tPBM is a relatively new research field, a dose–response curve has not yet been investigated or developed for most tPBM intervention protocols, particularly for the enhancement of cognition in healthy participants. A period of 4–8 weeks has been reported for tPBM to show significant effects in several human studies [18,19,57,58]. Consistently, our results demonstrated that a 4-week intervention period would be a workable and effective period through which to induce the significant effects of tPBM.

### 3.2. Justification of Pooling EEG Data during Rest and PVT Epochs in TP1 and TP2

In each experiment, we had a 10 min session (e.g., S1, S2, … S8) that consisted of two time periods (i.e., TP1 and TP2). Each TP included a 2 min rest and 3 min PVT epochs. Following the EEG data processing procedures, global (i.e., spatially averaged) nPSD_64_ values were quantified for each 2 min rest and 3 min PVT epoch within each TP1 and TP2 in the sham and tPBM groups in each of the 5-*f* bands. Paired t-tests were performed on paired nPSD_64_ values in each of 5-*f* bands between the rest and PVT epochs in each respective 5 min period in S1 for each (sham and tPBM) group. We found no significant difference in the nPSD_64_ values between these two epochs within TP1 or TP2 across both groups and five frequency bands. Thus, we pooled the EEG data during the 2 min rest and 3 min PVT epochs together in each of TP1 and TP2 for further data analysis.

### 3.3. Acute Effects and Topographies of Electrophysiology Induced by Initial LED-tPBM

To examine the acute effects of 10 min LED-tPBM, we first quantified and compared the global spectral values of nPSD_64_ between the sham and tPBM groups during TP1 and TP2 in Week 1, as shown in Figure 5a,b. These two figures clearly show a large difference in nPSD_64_ values between the two groups in the alpha frequency band (i.e., 8–13 Hz) during both TP1 and TP2 epochs. 

Next, the individual values of nPSD from the 64 channels were used to generate topographical maps for each frequency band for both TP1 and TP2, as shown in Figure 5c. Each of these maps was achieved using the EEGLAB extension in MATLAB. After careful statistical analysis, as described in Section 2.4.4, we observed significant enhancements in the normalized EEG powers in (1) the medial parieto-occipital regions in the delta band and (2) the left frontal, left temporal, and medial occipital regions in the alpha band. These significant increases occurred only during the last 5 min of the 10 min tPBM (i.e., TP2) with a maximal increase of up to 30% with respect to those in the sham group in several cortical regions.

### 3.4. Longitudinal Effects and Topographies of Electrophysiology Induced by 4-Week LED-tPBM

To investigate the longitudinal effects of the 4-week repeated LED-tPBM, the individual values of sham-subtracted nPSD (ss-nPSD) from the 64 channels were first calculated and used to generate topographical maps in each frequency band for both TP1 and TP2 at Week 4. Consequently, longitudinal effects were obtained by statistically comparing the ss-nPSD topographies in Week 1 with those in Week 4 for all 5-*f* bands during TP1 and TP2. The results are presented in Appendix A in Section SB of the Appendix A. The important lesson learned from this set of topographies was that the 4-week repeated LED-tPBM created widespread, significant increases in EEG spectral power across most scalp areas in all 5-*f* bands during TP2 with respect to those in the initial treatment in Week 1. Accordingly, we implemented an advanced image-processing algorithm that would facilitate improved spatial resolution, as presented below.

### 3.5. Extraction and Selection of gSVD Components as EEG Brain Networks

Following Step 5 (Figure 3), we selected 12 gSVD-derived PCs with their respective weights, 1D time series, and corresponding topographies. These extracted 12 gSVD components facilitated 2D topographies and 3D source localizations for the 12 EEG brain networks. Specifically, Figure 6a shows the diagonal values of the 64 gSVD components with their respective rankings based on their weights in the EEG signal after gSVD. As shown in this figure, an exponential decay in weight was observed across all components. Any component whose weight decayed by more than 90% when compared to the first/most-weighted component was excluded from further analysis. Accordingly, 12 dominant components were selected using this criterion (marked in red in Figure 6a). These 12 components contributed 73% of the entire EEG signal (i.e., the area under the curve of the 12 components divided by that of all 64 components) and served as the 12 EEG brain networks.

Next, a power spectral analysis was conducted on the time course of each gSVD component via the native MATLAB function “pwelch”. This resulted in a power spectral density (PSD) curve with a resolution of 0.05 Hz, ranging from 0 to 128 Hz for each gSVD component, subject, week, and TP (TP1 and TP2). Figure 6b illustrates an example of the group-averaged PSD curves from the sham (blue curve) and tPBM (red curve) groups during TP2 for component two (i.e., gSVD #2) after appropriate filtering.

### 3.6. Construction of 3D EEG Brain Networks Using eLORETA

After identifying the 12 components with gSVD, we considered them to represent or serve 12 EEG brain networks [36]. As these components were independent and orthogonal, they required minimal correlations. The Pearson Correlation Coefficient (PCC) was performed on each pair of 20 min (10 min for S1 and 10 min for S8) SVD components or networks for each subject to verify the least temporal correlations between all 12 gSVD components. Figure 6c depicts the group-averaged PCC values for every pair of the 12 networks. All the PCCs among these networks were less than 0.3, thus confirming the orthogonal and independent activities among those networks.

As each component had 64 dimensions (for the 64 electrodes in the spatial domain), they can be formed into a 2D topography of the relative electrical potential (rEP) in the sensor space. Accordingly, the 12 extracted brain networks (i.e., the 12 gSVD components) are shown in Figure 7 (generated in Step 5). To further compute the 3D models of the cortical current density of each identified EEG brain network, eLORETA was utilized to facilitate axial, sagittal, and coronal views of the current density of neural activity, as shown in the middle three columns of Figure 7. The 3D rendered brain templates’ top and side views of the left and right hemispheres are shown in the rightmost column of Figure 7. The yellow color on the rendered brain models indicates the binarized, associated cortical locations under a threshold of >75% of the maximum neural activity (i.e., cortical current density) in the network or 3D brain model. In other words, if the neural activity of the voxel lies within the top 25 percentile across all voxels in the brain model, the voxel was rendered yellow.

Using eLORETA, we identified key brain regions and cortical lobes with high neural activity (i.e., cortical electrical density) for each network, as listed in Table 2. For example, SVD #12 represents a network corresponding to the precentral gyrus and inferior parietal lobe, whereas SVD #1 and #3 reveal a network corresponding to the cingulate gyrus and precuneus.

### 3.7. Alterations of EEG Network Powers Induced by 4-Week LED-tPBM

Following Step 8 (see Figure 3), we calculated the EEG power changes in the 12 gSVD-derived brain networks that were induced by the 4-week tPBM in each of the five frequency bands from both groups in TP1 (Figure 8) and TP2 (Figure 9). Careful statistical analysis revealed that a 4-week tPBM significantly enhanced brain EEG powers in several networks across all five frequency bands compared to the sham stimulation. Moreover, such significant enhancements in network power occurred in more networks in TP2 than in TP1. Table 3 lists the number of EEG networks (i.e., the number of gSVD components) whose powers were significantly enhanced in TP1 and TP2 across the five frequency bands. The table clearly illustrates that Networks #9 and #11 were consistently more stimulated in the theta, alpha, and beta bands throughout the 10 min tPBM in S8 than in S1. Similarly, Networks #3, #5, and #6 were more power enhanced in the beta and gamma, theta and gamma, as well as theta and beta bands, respectively, throughout the tPBM period in S8 than in S1. Upon close inspection of the respective brain regions listed in Table 2, we concluded that 4-week LED-tPBM tended to increase the network power in (1) the frontal, parietal lobe, and right occipital lobe in the theta, alpha, and beta bands; (2) the limbic and parietal lobes in the beta and gamma bands; and (3) the bilateral frontal, parietal, and occipital lobes in the theta, beta, and gamma bands.

## 4. Discussion

Due to its low cost, ease of use, non-invasiveness, and low irradiance, LED-based tPBM is a preferable choice for neuromodulation. In this study, we conducted 64-channel EEG measurements from 22 healthy human subjects while they performed the PVT task concurrently with right-forehead LED-tPBM at the beginning and end of a 4-week intervention protocol. Behavioral improvement in vigilance/attention (i.e., reaction time) was recorded each time during the treatment for four weeks. Our results showed a gradual and significant improvement in the reaction time in the tPBM-treated group over four weeks. Meanwhile, we found that LED-tPBM had significant effects on EEG network power in several brain networks in the respective frequency bands. To the best of our knowledge, this is the first study to demonstrate the effects of LED stimulation on the longitudinal improvement of psychomotor vigilance, as well as the significant boost of electrophysiological functions in healthy subjects.

### 4.1. Effect of Repeated 4-Week LED-tPBM on Gradual Improvement of Psychomotor Vigilance

The longitudinal effects of repeated 4-week tPBM on reaction time are shown in Figure 4. These observations can be interpreted as follows. LED light delivery to the right forehead facilitates stimulation of metabolic activity in the electron transport chain of mitochondria, particularly in transmembrane protein complex IV (i.e., CCO). One of the major photoacceptors within the 600–1070 nm range of red/NIR light is CCO [1,2,3,59]. This increase in CCO activity provides more available neuronal metabolic energy (ATP) for performing tasks. An increase in ATP levels boosts cellular respiration, oxygenation, and hemodynamic activation [27,59]. Thus, with repeated tPBM over four weeks, LED stimulation improved attention by providing more energy in the form of ATP. This was reflected by the shortened reaction time in the tPBM-treated group. Furthermore, the reported observations demonstrated that significant improvement in psychomotor vigilance in healthy humans can be achieved by (1) repeated right-forehead tPBM over 4 weeks, (2) a low irradiance of LED clusters (if weekly routines of tPBM are carried out), and (3) a simple and sparse intervention schedule of 10 min per treatment and twice per week.

### 4.2. Acute EEG Power Enhancement of Brain Oscillations by LED-tPBM

In this study, we observed that LED-tPBM in S1 at Week 1 enabled acute increases in EEG nPSD in the left frontal, left temporal, and medial occipital regions in the alpha band (Figure 5c). Cognitive stimulation and/or enhancement have been reported to be associated with an alpha power increase in the frontal and parietal regions [60,61,62]. Furthermore, the prefrontal cortex is known to play a key role in sustained attention, is activated in response to tasks requiring vigilance/attention [63], and controls the executing function related to improving the reaction time during the task [64]. The salience network is also said to be present in the prefrontal cortex [65], which is responsible for the detection of the distinct target inputs that are involved during PVT. The ability of LED-tPBM to neuromodulate the electrophysiological activity of the frontal cortex may account for the gradual and significant improvement in psychomotor vigilance in the tPBM-treated group.

An increased nPSD value in the parietal lobe was also observed under LED stimulation in the delta and alpha bands (Figure 5c). The parietal lobe is said to modulate top-down attention, thus allowing for faster reactions toward the stimulus [66,67,68]. Several studies have shown that the lateral frontal area and inferior parietal lobe together support executive control and attention maintenance [69,70,71]. In particular, the left inferior parietal lobe has been shown to be activated in subjects with a faster reaction time [66,67]. Consequently, the increased EEG power in the inferior parietal lobe by LED-tPBM could explain the faster reaction time in the tPBM-treated subjects over time.

Furthermore, significant enhancement in the EEG nPSD by LED-tPBM on the right forehead occurred contralaterally. Such contralateral effects of tPBM have been reported by our group and others. The underlying mechanisms need to be explored further in future studies.

### 4.3. EEG Power Improvement Globally in Theta, Beta, and Gamma Bands by Repeated LED-PBM

In this study, we observed that four weeks of LED-tPBM increased EEG nPSD globally in the theta, beta, and gamma bands (Appendix A in the Appendix A). It has been reported that the attention and efficient processing of the task are often associated with theta [72,73], beta [73], and gamma bands [74,75,76,77]. First, the theta band is often associated with the efficient processing of cognitive tasks and attention [72,73,78]. An increase in theta activity in the frontal areas indicates its involvement in working memory tasks [79,80,81]. Previous studies have revealed that an increase in theta and gamma power represents a neural correlation of working memory processing [82,83,84].

Second, beta oscillations are associated with cognitive functions, such as working memory [85,86,87,88], the executive control of action [89,90,91], state of alert [73], and the prevention of distraction [92,93]. Studies have suggested that beta oscillations convey moment-to-moment top-down modulatory signals to the lower sensory cortices, thus maintaining existing mental states [94,95]. Wrobel et al. showed that beta band activity reflects arousal of the visual system during visual attention tasks [96]. Beta oscillatory responses have been considered to be related to motor and somatosensory functions [97].

Finally, gamma oscillations have been extensively studied in cognitive processes engaged in attention, memory, and perception [84,98,99,100,101,102], as well as correlated with memory loading, formation, and maintenance [103,104,105]. Gamma oscillations have been proposed to play a role in the mechanisms of synaptic plasticity and memory formation [101,102,106], as well as been shown to be modulated by the cognitive processes engaged in spatial working memory in monkeys [86] and recognition memory tasks in humans [107].

Consequently, continuous and repeated intervention of LED-tPBM on the electrophysiological activities of the human brain over four weeks provided longitudinal stimulations in EEG power in those brain regions, which can explain the gradual and significant improvement in psychomotor vigilance in the tPBM-treated group.

### 4.4. Large-Scale Neural Activities Presented by gSVD-Derived EEG Brain Networks

#### 4.4.1. Similarity of gSVD-Derived EEG Brain Networks to fMRI-Defined Networks

The application of gSVD with eLORETA enabled us to isolate and identify 12 intrinsic EEG brain networks, as shown in Figure 7, which accounted for 73% of the total contribution to the recorded EEG signal. This is consistent with the neural physiology that communication among neurons and functional activity in the human brain takes about 60–80% of the total energy [108]. We recognized spatial co-localizations between the gSVD-identified EEG networks and fMRI-recognized networks [109,110,111,112,113,114] by inspecting the active cortical localizations of each EEG network, as shown in Figure 7 and Table 2. Although EEG records electrophysiological oscillations at much higher frequencies with a lower spatial resolution than fMRI, gSVD-derived EEG brain networks may reflect the neural activity resulting from those identified by fMRI-defined brain networks.

The first seven most-weighted gSVD-derived EEG networks accounted for approximately 61% of the total EEG signal contribution. Networks #1, 2, 3, and 8 were found to be associated with the DMN, which is considered the most important and dominant network in the human brain [115,116,117]. Our results are consistent with this statement. Specifically, the networks that are associated with the DMN accounted for almost 43.6% and 60% of all EEG signals (64 components) and the 12 most-weighted/dominant brain networks, respectively. Thus, gSVD, along with eLORETA, offers a potentially feasible means of extracting the dominant DMN fluctuations of the human brain from EEG recordings. The left and right frontoparietal networks (L- and R-FPN) and executive control networks (ECN) were also identified with the gSVD algorithm and were ranked as #4–7. These networks contributed to approximately 27.5% of the 12 dominant networks. The FPN is a network commonly observed during sustained attentional tasks [39]. The FPN is known to play an important role in cognition [114,118,119], attention, memory consolidation, and memory encoding [120,121,122,123]. Similarly, the ECN plays a crucial role in human executive control functions during sustained attentional tasks such as cognitive inhibition, inhibitory control, working memory, reasoning, memory, problem solving, and planning [120,124,125]. In short, the ability of the gSVD algorithm along with eLORETA to identify fMRI-defined, cognition-sensitive networks (i.e., DMN, L-FPN, R-FPN, and ECN) based on measured EEG signals would expand the applications of EEG as a potential neural monitoring tool, along with a high temporal resolution.

#### 4.4.2. Enhancement of Network Power by Repeated LED-tPBM in Selected Brain Networks

As listed in Table 3, the tPBM-treated group showed increased network power in several brain networks when compared with the sham group. First, Networks #9 and #11 were consistently more stimulated in the theta, alpha, and beta bands after four weeks of repeated LED-tPBM. These two networks include the cortical regions of the precentral gyrus, postcentral gyrus, inferior parietal lobule, right middle occipital gyrus, and right cuneus. All these brain regions are associated with cognitive function in humans. For example, the inferior frontal and parietal lobules play a major role in decision making [126,127]; the parietal lobule is involved in information manipulation during working memory [128]; and the occipital lobe is required for perception [128]. Hence, we speculated that the LED-based tPBM induced the augmentation of vigilance as a result of the enhancement of network power in these cognition-related EEG cortical networks.

Meanwhile, Networks #3, #5, and #6 were more power enhanced in the respective frequency bands (see Table 3). These networks cover the brain regions of the cingulate gyrus, precuneus, right and left inferior frontal and parietal lobules, and precuneus. These regions are closely linked to human cognitive functions. In particular, the increase in beta power in the parietal regions for faster reaction times has been related to attention processing in many fMRI studies [129,130,131]. In this study, the power enhancement of the parietal lobule could be related to the enhancement of the parietal attention network. It is also possible that beta band enhancement in the frontoparietal regions reflects top-down attentional control [132]. These results agree with a previous MEG study showing that beta oscillations serve as a mechanism for spreading attentional arousal among higher cortical areas [133]. Thus, the increase in beta band activity in all cortical areas could potentially explain the improvement in vigilance.

Finally, it is worthwhile to summarize and emphasize that our newly developed gSVD algorithm combined with eLORETA facilitates (1) the identification of large-scale EEG brain networks, many of which are consistent with those found from fMRI, and (2) the quantification of enhanced network power in the respective brain networks by 4-week repeated LED-tPBM.

### 4.5. Limitation of the Study and Future Work

This study has several limitations. First, the sample size was small; therefore, statistical power was limited. In particular, we were not able to perform a comprehensive and exhaustive statistical analysis. Second, the LEDs used in the study were two types of LED clusters at 660 nm and 810 nm, which prevented us from determining the exact contributions of the two light wavelengths to the significant effects of LED-tPBM. Third, the experimental design was unnecessarily complex and included two blocks of rest and a PVT task, which may cause confusion and difficulty for the reader. Finally, the EEG data were collected only during the tPBM interventions; no post-tPBM measurements were taken. Accordingly, further studies should include a larger sample size to warrant excellent statistical power; they should use a single-wavelength (e.g., 810-nm LED) light source with a simpler experimental design and include post-tPBM assessments.

## 5. Conclusions

This sham-controlled study was a proof-of-principle methodology development investigation. It provides substantial evidence that supports or confirms our three hypotheses: (1) repeated LED-tPBM of the right forehead of healthy subjects significantly improves psychomotor vigilance; (2) acute LED-tPBM significantly increases topographical EEG delta and alpha powers; and (3) the four-week repeated LED-tPBM enhances several brain network strengths (such as DMN, FPN, and ECN) in all five EEG frequency bands when compared to sham stimulation. Specifically, we showed, for the first time, acute increases in EEG delta and alpha powers during 10 min LED-tPBM while participants performed the PVT task. We also demonstrated that theta, beta, and gamma EEG powers significantly increased globally in most of cortical regions after four weeks of LED-tPBM. With a novel analysis of combining gSVD with eLORETA, we enabled the identification of 12 independent and orthogonal brain networks in both sensor and brain space, as well as found that these networks matched well with several fMRI-recognized brain networks. Our results clearly suggest that a 4-week prefrontal LED-tPBM can neuromodulate the DMN, FPN, and ECN, which may indicate a possible causal effect between modulated brain networks and improved psychomotor vigilance outcomes. Future work should include more participants for stronger statistical power and should include a single-wavelength light source and a simpler experimental design.

## Figures and Tables

**Figure 1 bioengineering-10-01043-f001:**
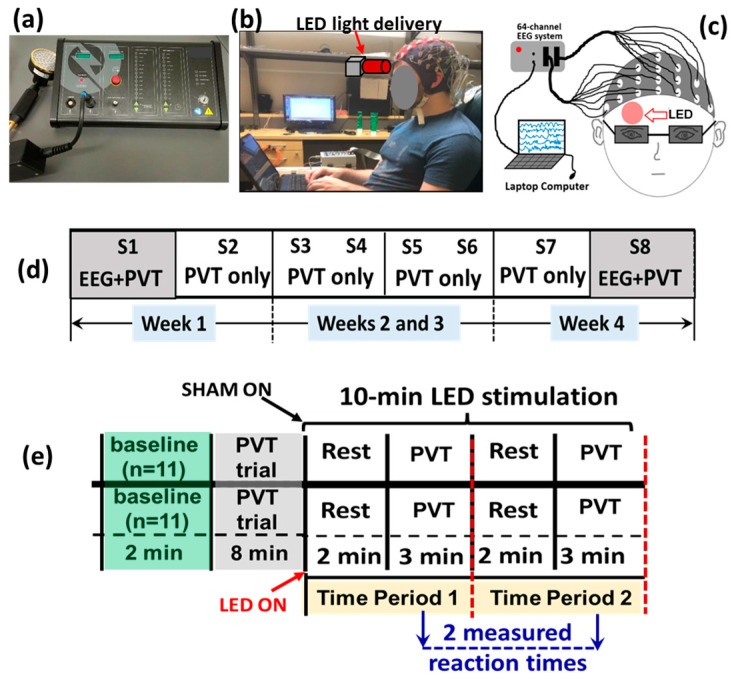
A photograph of (**a**) the LED-tPBM unit used for the study, and (**b**) a participant wearing an EEG cap and performing PVT while the LED light was delivered to (**c**) the right forehead. (**d**) Diagram of the experimental protocol showing a total of eight sessions of the experiment, S1 to S8, over four weeks. EEG measurements were taken concurrently while the participants performed PVT only in S1 of Week 1 and S8 of Week 4. (**e**) A schematic illustration for each of the eight sessions. This between-group experimental design included 11 subjects randomly assigned to either the active or the sham group. The entire experiment lasted for 20 min. During the first ten minutes, the participants in both groups were not subjected to any treatment. During the last ten minutes, the active (n = 11) or sham (n = 11) group was administered with true tPBM or sham treatment, respectively, on the right forehead. One 2 min resting epoch and 3 min PVT epoch together consisted of one time period (TP). Participants in each group underwent TP1 and TP2 at each visit. PVT was performed twice during the 10 min tPBM period per visit. The respective reaction times were recorded twice per visit, resulting in a total of 16 reaction times measured over 4 weeks (2/visit × 2 visit/week × 4 weeks = 16).

**Figure 2 bioengineering-10-01043-f002:**
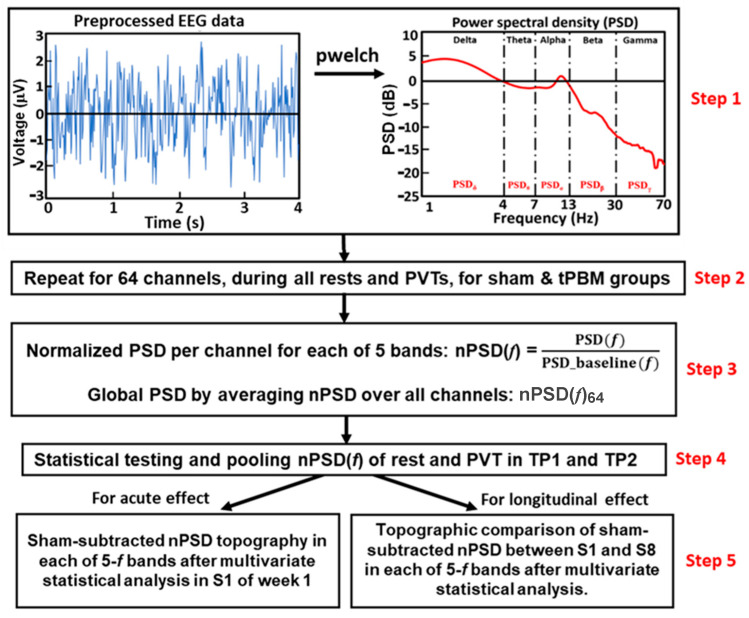
A flow chart showing the five steps for (1) converting a time-domain series to a frequency-domain PSD, namely, PSD(*f*), where f covers five EEG frequency bands, (2) repeating Step 1 for all the channels for each of the four rest/PVT epochs in the two separate groups, (3) calculating the channel-wise and channel-averaged (global) normalized PSD(*f*), (nPSD(*f*) and nPSD(*f*)_64_), (4) statistical testing on whether the global nPSD(*f*) values were significantly different between the rest and PVT epochs in each of TP1 and TP2, and (5) regrouping nPSD to cover the 5 min TP1 and TP2 for each channel in each of five frequency (5-*f*) bands, as well as constructing sham-subtracted nPSD topography for each of 5-f bands after multivariate statistical analysis for the TP1 and TP2 during the 10 min tPBM/sham epoch. This analysis was performed separately to examine the acute effect during Session S1 (left panel), and to determine the repeated effect of the 4-week LED-tPBM by comparing the nPSD values between S1 and S8 (right panel).

**Figure 3 bioengineering-10-01043-f003:**
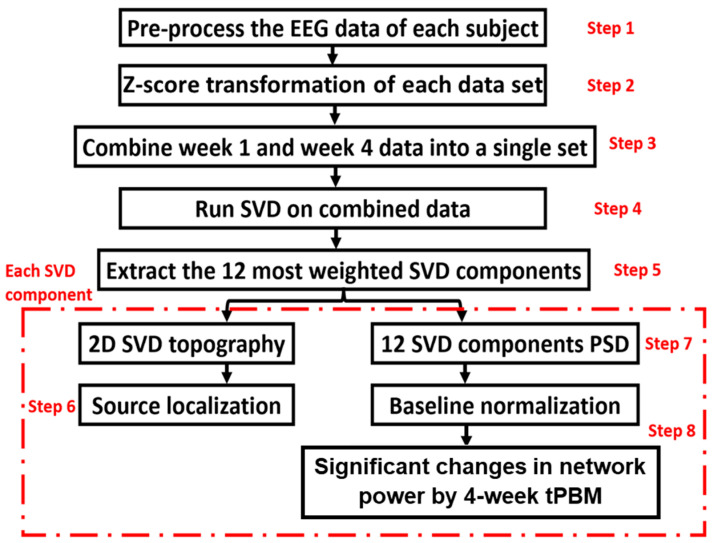
An EEG data processing flowchart showing the eight steps through which to obtain 2D- and 3D-brain EEG networks, as well as the respective alterations induced by 4-week LED-tPBM. All steps were performed using MATLAB except Step 6, which was performed using eLORETA software. See text and Section SA of the Appendix A for details on each processing step.

**Figure 4 bioengineering-10-01043-f004:**
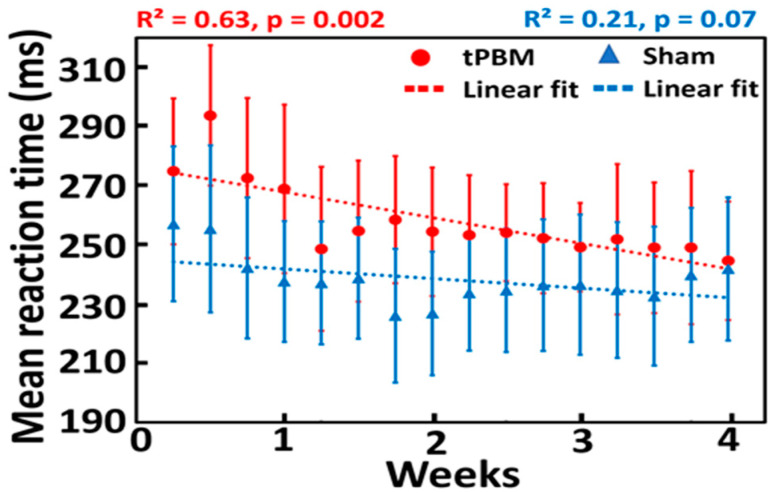
Effect of repeated 10 min LED-tPBM on the reaction time for the sham (n = 11; blue) and tPBM (n = 11; red) groups. Both dotted lines are linear fits to the sham (blue) and tPBM (red) groups, with *p*-values of 0.07 and 0.002, respectively. Each data point reflects the mean reaction time averaged over each respective group during the 3 min PVT in each session from S1 to S8 over four weeks. Note that, in each session (e.g., S1), two reaction times were recorded in TP1 and TP2, leading to a total of 16 data points over 4 weeks.

**Figure 5 bioengineering-10-01043-f005:**
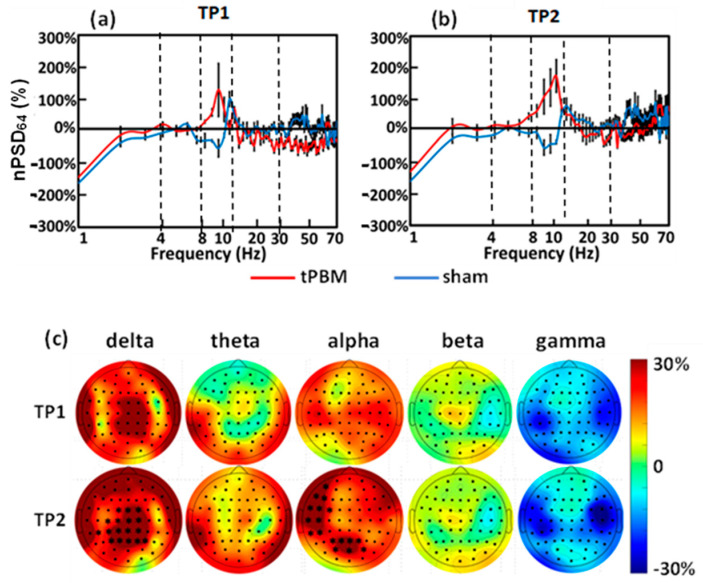
Normalized EEG power spectrum density plots averaged over all 64 channels (i.e., nPSD_64_) from the sham (blue curve) and tPBM (red curve) groups during (**a**) TP1 and (**b**) TP2. Normalization was calculated with respect to the 2 min baseline taken at the beginning of the experiment without light stimulation (see Figure 1e). The x-axis denotes the EEG frequency in Hz, and the y-axis represents the percentage change in nPSD_64_ relative to the PSD_64_ during the 2 min baseline. (**c**) Topographic maps of the nPSD at each of 5-*f* bands. This also shows the statistical comparisons of the channel-interpolated topographical alterations in the nPSD between the sham (n = 9) and tPBM (n = 7) groups during TP1 and TP2 in Week 1. The columns indicate the delta, theta, alpha, beta, and gamma bands. The color bar indicates the percentage change in the nPSD with respect to the baseline for each electrode channel. The “*” marks a *p* value < 0.05 for statistical significance of nPSD between tPBM and sham groups after multivariate comparison correction.

**Figure 6 bioengineering-10-01043-f006:**
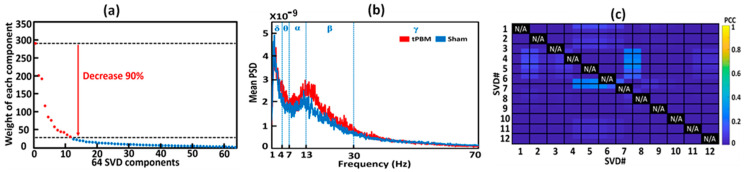
(**a**) Ranks and weights of all the 64 SVD components after gSVD. The x-axis indicates the ranks of the components, while the y-axis denotes the weight of each component. The 12 most-weighted components are marked by red dots and selected for further data processing. The total weight of the 12 most-weighted components (red dots) takes 70% of the total weight (all the dots). (**b**) The group-averaged PSD curves from the sham (blue) and tPBM (red) groups during TP2 for component two (SVD #2). Blue vertical lines mark the five EEG frequency bands; namely, delta (δ: 0.5–4 Hz), theta (θ: 4–7 Hz), alpha (α: 7–13 Hz), beta (β: 13–30 Hz), and gamma (γ: 30–70 Hz). (**c**) A correlation matrix to show the group-averaged PCC values for every pair of the 12 networks. The vertical and horizontal axis depict the 12 gSVD components, and the color bar represents the PCC between each pair of networks. Since all the self-correlations were meaningless, all the values along the diagonal line were marked as N/A (i.e., not applicable).

**Figure 7 bioengineering-10-01043-f007:**
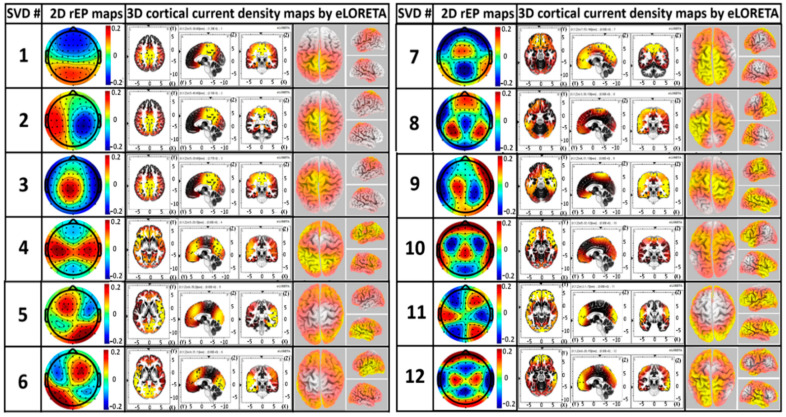
The 2D rEP maps of the 12 identified gSVD components (labeled by SVD#) with 3D source localizations of cortical current density. For each of the two panels, the second left most column shows the 2D rEP map; the middle three columns reveal the axial, sagittal, and coronial views for each rEP map, with a unit of cm in all three dimensions; and the two right most columns depict the top and side views of the left and right hemispheres for the 3D rendered brain templates. The yellow color in each brain model indicates the binarized cortical current density for a threshold of >75% of the maximum neuronal activity in the respective brain models. The color bars show the rEP of the ‘dipole’ across the scalp (no unit).

**Figure 8 bioengineering-10-01043-f008:**
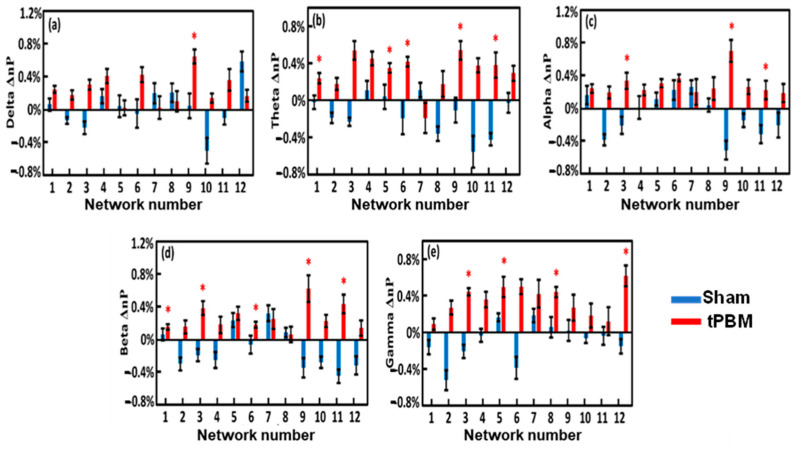
Group-level EEG power changes, Δnp, between S8 (in Week 4) and S1 (in Week 1) for the 12 brain networks in (**a**) delta, (**b**) theta, (**c**) alpha, (**d**) beta, and (**e**) gamma bands during TP1 from the sham (blue) and tPBM (red) groups. The standard error of the mean is represented by the error bars. The significant differences of Δnp between the two groups were determined using the two-sample non-parametric test at the significance level of *p* < 0.05 and are indicated by ‘*’.

**Figure 9 bioengineering-10-01043-f009:**
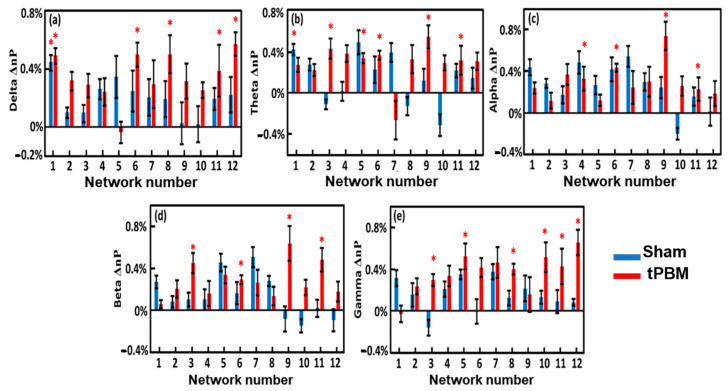
Group-level EEG power changes, Δnp, between S8 (in Week 4) and S1 (in Week 1) for the 12 brain networks in the (**a**) delta, (**b**) theta, (**c**) alpha, (**d**) beta, and (**e**) gamma bands during TP2 from the sham (blue) and tPBM (red) groups. The standard error of the mean is represented by the error bars. The significant differences of Δnp between the two groups were determined using the two-sample non-parametric test at the significance level of *p* < 0.05 and are indicated by ‘*’.

**Table 1 bioengineering-10-01043-t001:** LED-tPBM parameters used in the study.

Wavelength	660 and 810 nm
Operation model	CW
Aperture diameter	6.4 cm
Total aperture area	32 cm^2^
Total number of LED emitters	69
Averaged irradiance	~30 mW/cm^2^

**Table 2 bioengineering-10-01043-t002:** Main associated cortical lobes and regions for the 12 networks that were determined using eLORETA. (# marks the number of gSVD components).

gSVD	Associated Cerebral Lobes	Associated Brain Regions
# 1	Limbic lobe, parietal lobe	Cingulate gyrus, precuneus
# 2	Limbic lobe, parietal lobe	Cingulate gyrus
# 3	Limbic lobe, parietal lobe	Cingulate gyrus, precuneus
# 4	Left: frontal, parietal lobe	Left: inferior frontal gyrus, inferior parietal lobule
# 5	Left: frontal, parietal, occipital lobe	Left: inferior frontal gyrus, inferior parietal lobule, precuneus
# 6	Right: frontal, parietal, occipital lobe	Right: inferior frontal gyrus, inferior parietal lobule, precuneus
# 7	Medical frontal lobe, limbic lobe	Medial frontal gyrus, anterior cingulate, cingulate gyrus
# 8	Frontal lobe, temporal lobe	Cingulate gyrus, lateral visual lobule
# 9	Frontal lobe, parietal lobe	Precentral gyrus, postcentral gyrus, inferior parietal lobule
# 10	Frontal lobe, limbic lobe	Cingulate gyrus
# 11	Right: occipital lobe	Right: middle occipital gyrus, cuneus
# 12	Frontal lobe, parietal lobe	Precentral gyrus, inferior parietal lobe

**Table 3 bioengineering-10-01043-t003:** The EEG networks (i.e., gSVD components) that were stimulated by 4-week tPBM.

Frequency Band	TP1	TP2	Common Networks in TP1 and TP2
delta	9	1 6 8 11 12	
theta	1 5 6 9 11	1 3 5 6 9 11	1 5 6 **9** **11**
alpha	3 9 11	4 6 9 11	**9** **11**
beta	1 3 6 9 11	3 6 9 11	3 6 **9** **11**
gamma	3 5 8 12	3 5 8 10 11 12	3 5 8 12

Note: The network numbers written in red indicate new networks that only appeared in TP2, not in TP1. TP1: time period 1; TP2: time period 2. Bolded numbers indicate the networks that appeared three times.

## Data Availability

The data presented in this study are available on request from the corresponding authors.

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
