# Peer review of "Repeated Transcranial Photobiomodulation with Light-Emitting Diodes Improves Psychomotor Vigilance and EEG Networks of the Human Brain"

_bioengineering, 2023, doi:10.3390/bioengineering10091043_

Round 1

Reviewer 1 Report

The paper by Chaudhari et al, examined the effect of repeated transcranial photobiomodulation with LED on viglilance and on EEG spectral and topographical characteristics. Transcranial LED photobiomodulation (tPBM) is a new topic that the scientific literature has recently begun to deal with. There are few papers on tPBM and EEG in the literature, so Caudhari's work is certainly interesting.

However, there are many methodological aspects that need to be clarified and some points of the discussion that need to be explored.

·        Even if the recruitment involved 11 subjects per group, at a certain point it is clear from the text that the subjects actually used for the EEG analysis are less. This certainly has a very strong impact on the choice of statistical tests (normality trst, effectively compared subjects...); moreover, the statistical tests are not reported in an exhaustive manner, reporting degrees of freedom and test value. This imbalance also has consequences in step 3 of paragraph 2.4.5.

·        Why the authors did not compare the reaction times between the two groups, but only their progress through the various sessions. Figure 4 shows a marked difference between groups: how do the authors justify it?

·        It is better to explain the choice of the 12 gSVD components in the methods and not in the results.

·        All the details on the images used for eLORETA are missing.

·        Why do the authors use the "pwelch" function on epochs of different lengths in the case of EEG signals (4 seconds with 50% overlap) and SVD components (20 seconds with 50% overlap)?

·        The length of rest (2 minutes) and PVT (3 minutes) is different, resulting in different sample size for PSD.

·        It is not clear what the "sham-subctration" shown in paragraph 2.4.4. means?

·        It is absolutely not justifiable to merge data acquired under rest conditions and during tPBM as stated in paragraph 3.2. as functionally different. Furthermore, shortly after, the authors claim to normalize the data with respect to the 2-minute baseline (see legend in figure 5).

·        The discussion broadly addresses the topographical aspects of the results, but less convincingly the link between the increase in power and the involvement of the areas. To give an example: during the execution of a movement, the decrease in power in the beta band in the central regions coincides with the active involvement of that area. Paragraphs 4.2 and 4.3 should be made clearer.

Minor points:

·        Which kind of timer is used in the PVT?

·        The choice of the position of the stimulator on the right forehead is not well illustrated.

·        From the second line of paragraph 2.4.2. it seems that the EEGs acquired during the tPBM were analysed.

·        Localization of EEG sources allows identification of cortical, non-subcortical components (paragraph 3.6.).

·        In the discussion (paragraph 4.2) the authors should account for activation of areas on the left versus stimulation on the right.

Reviewer 2 Report

In this study, Transcranial photobiomodulation (tPBM) has been suggested as a non-invasive neuro-modulation tool. Repetitive administration of light-emitting diode (LED)-based tPBM for several weeks significantly improved human cognition. To understand the electrophysiological effects of LED-tPBM on the human brain, we investigated alterations by repeated tPBM in vigilance performance and brain networks using electroencephalography (EEG) in healthy participants.The review comments are as follows:

1.         Figure 1:It is suggested that the author adjust the distribution position and size of the four pictures, especially Figure a and Figure c.

2.         Figure 6:It is suggested that the author adjust the distribution position and size of the three pictures.

3.         Figure7:It is suggested that the author adjust the distribution position and size of the three pictures.

Author Response

Please see the attached PDF file.

Reviewer 3 Report

In this paper, authors investigated the LED-based transcranial photobiomodulation on the human brain. By using LED-tPBM stimulation delivered to the right forehead using a continuous-wave LED handheld probe, and a 64-electrode EEG system signals were recorded from each participant. It is interesting that that theta, beta, and gamma EEG powers significantly increased globally after four weeks of LED-tPBM. In my opinion, this study and obtained results are attractive. There are only some minor comments for improving the paper as follows.

a) I do not understand sentences in lines 110 and 112: On the other hand, the power used for sham was set to be 0.0 W. On the other hand, the power used for sham was set to be 0.0 W.

b) I wonder about the length of time (the 4-week) in the experimental protocol. Is it an optimized period?

c) Please emphasize further the advantages of your new algorithm to identify EEG networks.

Round 2

Reviewer 1 Report

Thanks to the authors for the detailed responses.

I still have some doubts about the merge of data at rest and during a task, although they are superimposable from a spectral point of view. I would have liked the authors to justify the total superimposability of EEG data at rest and during task also with literature data.

Alternatively, the authors should more clearly restate the nature of the study as a "proof-of-principle methodology-development study" or "pilot study", both in the introduction and in the conclusions.

Author Response

We thank Reviewer 1 for the careful and helpful suggestion. We have added the requested statement in the Introduction and Conclusion Sections. The changes are written in red and Bold font. Please check Lines 71 and 660.